# Verification of a Deep Learning-Based Tree Species Identification Model Using Images of Broadleaf and Coniferous Tree Leaves

Yasushi Minowa [1,*], Yuhsuke Kubota [2] and Shun Nakatsukasa [1]

1 Graduate School of Life and Environmental Sciences, Kyoto Prefectural University, Kyoto 6068522, Japan; nakatsushun@gmail.com
2 Faculty of Life and Environmental Sciences, Kyoto Prefectural University, Kyoto 6068522, Japan; terayu1819@gmail.com
* Correspondence: sharmy@uf.kpu.ac.jp; Tel.: +81-75-703-5684

**Abstract:** The objective of this study was to verify the accuracy of tree species identification using deep learning with leaf images of broadleaf and coniferous trees in outdoor photographs. For each of 12 broadleaf and eight coniferous tree species, we acquired 300 photographs of leaves and used those to produce 72,000 256 × 256-pixel images. We used Caffe as the deep learning framework and AlexNet and GoogLeNet as the deep learning algorithms. We constructed four learning models that combined two learning patterns: one for individual classification of 20 species and the other for two-group classification (broadleaf vs. coniferous trees), with and without data augmentation, respectively. The performance of the proposed model was evaluated according to the MCC and F-score. Both classification models exhibited very high accuracy for all learning patterns; the highest MCC was 0.997 for GoogLeNet with data augmentation. The classification accuracy was higher for broadleaf trees when the model was trained using broadleaf only; for coniferous trees, the classification accuracy was higher when the model was trained using both tree types simultaneously than when it was trained using coniferous trees only.

**Keywords:** AlexNet; broadleaf trees; Caffe; coniferous trees; deep learning; F-score; GoogLeNet; MCC; tree species identification



## 1. Introduction

We developed an auto-tree-identification system based on leaf images for mobile terminals [1–6]. In previous studies, we focused mainly on woody plant species and investigated whether tree species could be identified from leaf images. In tree species identification using leaf images, images are captured in a manner such that only a single leaf is included. Pre-processing (e.g., image processing) is then performed. Various information from the images is used as identification criteria; this information includes shape features [7–11], texture [12,13] and leaf venation [14–16]. Supervised learning with machine learning methods [17], such as neural networks and decision trees, is used to discriminate among candidate species [18]. Minowa et al. [1] did not achieve high classification accuracy when they applied machine learning to shape features. Minowa et al. [2] demonstrated that classification accuracy could be improved by combining shape and leaf venation information. However, while classification accuracy was high for training data, performance was poor for test data. Several of our subsequent studies [4–6] demonstrated that deep learning can be applied to tree species identification with high classification accuracy, even for test data.

Deep learning refers to a neural network model that consists of multiple layers, such as convolutional neural networks (CNNs); such models have recently achieved excellent results with respect to image recognition. Prior to the emergence of deep learning, most

image recognition methods generally used pre-extracted image features. The accuracy of image recognition is affected by the type of features used. However, image features are difficult to specify because they vary among objects. Thus, the extraction of image features is greatly influenced by researcher and developer experience [19]. Deep learning does not require users to define image features in advance. The user inputs a large amount of training data into the computer, which then extracts image features from the training data to perform recognition. Most deep learning applications are easy to use, publicly available and free of charge, which facilitates their use by researchers in various fields; thus, people in fields other than machine learning can conduct research using deep learning. With regard to tree species identification using leaf images, deep learning enables the omission of complicated image processing performed on the leaves for analysis. Accordingly, the analyst is not required to process numerous leaf images using image analysis software to extract shape features and venation information from a single leaf image.

State-of-the-art deep learning algorithms have been proposed at large-scale international competitions for fields related to image recognition. These include ILSVRC [20], COCO Challenge [21], ImageCLEF/LifeCLEF [22] and Places2 Challenge [23]. Among these competitions, LifeCLEF includes a sub-competition called PlantCLEF, in which participants strive to achieve high classification accuracy for various types of plant datasets [17]. In the 2015 PlantCLEF competition, plants were divided into seven parts (branch, entire plant, leaf [photographic image], leaf [scan or scan-like image], flower, fruit and stem) to show competitive classification accuracy. The 2015 PlantCLEF competition included 113,205 images of 1000 species of herbs, trees and ferns, mostly from France. The GoogLeNet model had the highest overall classification accuracy, but its mean reciprocal rank was only 0.667 (i.e., not high). Ghazi et al. [17] applied transfer learning [24] and other techniques to the AlexNet, GoogLeNet and VGGNet deep learning models, using the same dataset as the 2015 PlantCLEF competition; they achieved high classification accuracies with leaf scans and scan-like images but lower classification accuracies with leaf photographs. Because ImageCLEF and PlantCLEF focus on competition with deep learning algorithms, they do not extensively explore the tendency to misclassify targets. We speculated that the low classification accuracy of the 2015 PlantCLEF was partly related to the large number of plant species used for identification. Deep learning has been used to assess the extent of plant diseases [25,26], identify tree species based on Lidar and drone-acquired images [27–29] and to distinguish broadleaf and conifer species [30–34]. Most such studies use scanned images of single leaves rather than multiple leaves or images from databases such as PlantCLEF2012 [35] or the Swedish leaf dataset [36], whereas few studies have directly used images of multiple leaves.

In our previous studies, tree species were broadly classified into broadleaf [5] and coniferous [6] leaf images; deep learning was applied to tree species identification. The main objective of those studies was to determine whether deep learning could be used to identify tree species based on photographs of multiple leaves that were acquired outdoors. First, we conducted tree species identification for broadleaf trees using machine learning [5]. Second, we conducted tree species identification for coniferous trees based on the results of broadleaf tree analyses [6]. Therefore, our constructed deep learning model could not be applied to datasets that contain both broadleaf and coniferous trees. However, on the basis of tree species identification using machine learning and deep learning, we aim to develop a system that can perform tree species identification for numerous users using various mobile terminals in the future. For users who are familiar with tree species identification, it is appropriate to construct a tree species identification system that is adjusted for broadleaf and coniferous trees in advance. For users who are unfamiliar with tree species, the species of a tree should be automatically identified by simply acquiring a photograph of its leaves.

The objective of this study was to verify the accuracy of tree species identification using deep learning with broadleaf and coniferous leaf images acquired in the field (except for instances in which broadleaf and coniferous images are simultaneously reflected in a single image). We used 12 broadleaf and eight coniferous trees. Thus, we calculated the

accuracy of tree species identification when 20 species were divided into two groups: all 12 broadleaf trees were in the "broadleaf group" and all eight coniferous trees were in the "coniferous group". We then compared the initial accuracy with the accuracy of tree species identification when only broadleaf trees [5] and only coniferous trees [6] were used. Some deep learning models, such as pipelined and linked models, have functions that allow multiple deep learning models to be connected and regarded as a single model [37,38]. The size (capacity) of a learning model is especially important when the model is implemented in a mobile terminal, but the above functions enable construction of a model that is much smaller than conventional models. Moreover, if separately trained models can be combined, broadleaf and coniferous trees can be classified in advance and then identified using their respectively trained models. Overall, we focused on determining the accuracy of tree species identification when all tree species are classified simultaneously (i.e., without distinguishing between broadleaf and coniferous trees), compared to the accuracy when two groups of trees are classified separately.

## 2. Materials and Methods

### 2.1. Study Sites

Leaf images of 12 broadleaf tree species [5] and 8 coniferous tree species [6] were used for analysis at the Kyoto Prefectural University campus and Kyoto Botanical Gardens (Figure 1). All broadleaf tree species had simple leaves but were classified into three different categories: simple leaves with smooth margins, simple leaves with toothed margins and lobed leaves. We photographed 300 leaves from each tree. Previous studies of broadleaf trees [2–5] used these same leaf shapes, which facilitated comparison with our results.

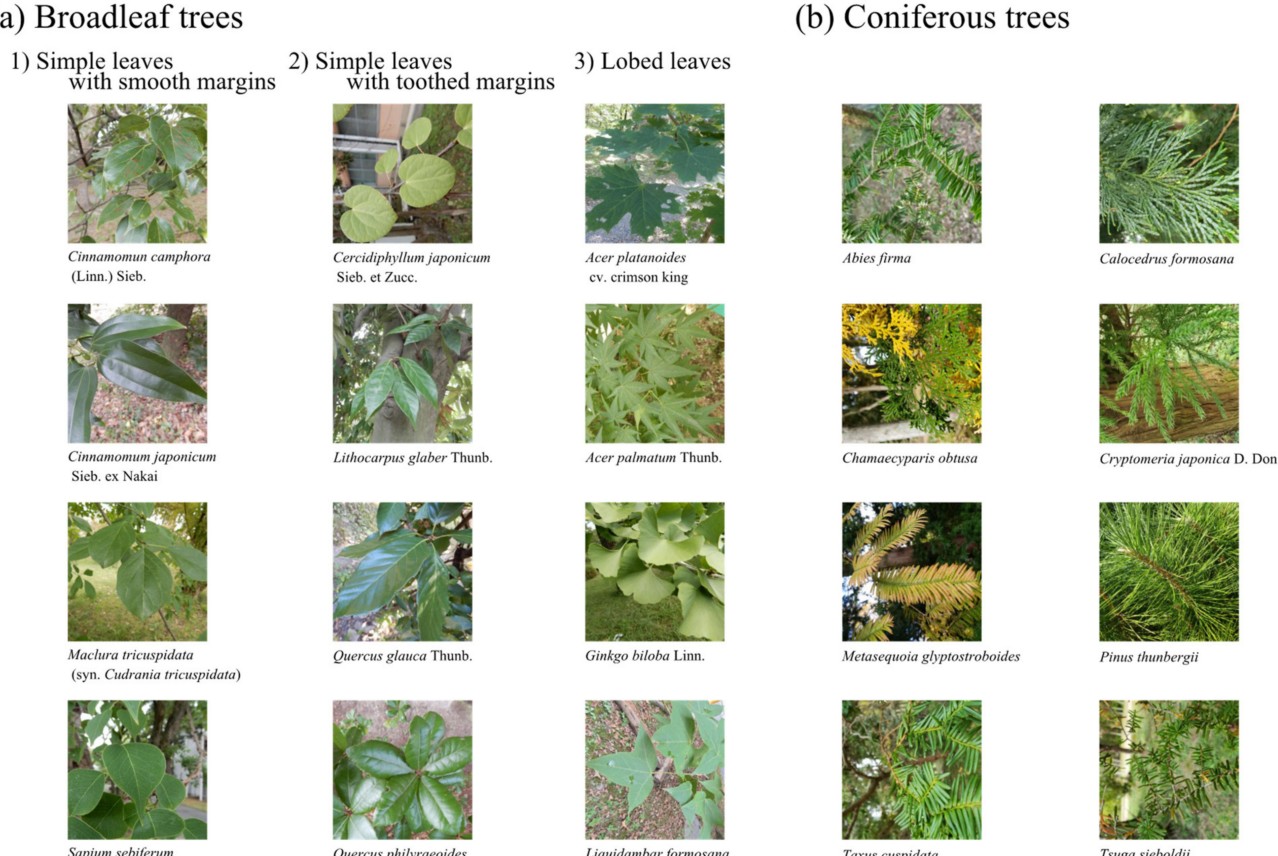

**Figure 1.** Focal species and classification based on broadleaf and coniferous trees.

### 2.2. Photographic Methods

Because broadleaf and coniferous trees were photographed by different photographers on different dates, a NICON COOLPIX A900 [39] was used for broadleaf trees and a Galaxy S9 was used for coniferous trees [40]. Photographs were acquired from September 2020 to November 2020. When possible, photographs were acquired on sunny days under calm conditions to ensure consistent image color tone. The distance between the subject and the camera was approximately 0.3–1.0 m. All images were in color. We did not process additional (i.e., non-leaf) elements, such as the sky, ground or buildings, which were present in the background of some photographs. We acquired photographs from various angles to avoid capturing the same leaf when possible; to ensure that the leaves were generally centered, we did not enlarge images. In deep learning, background objects are important. Since the distance between the subject and the camera was approximately 0.3–1.0 m, in many cases the background is barely visible in the image in this study. If the background objects are reflected in the image, we acquired leaf images that were taken at various angles so that the background of the leaf image would be various background images, such as the sky, ground (soil, concrete), buildings and other structures.

### 2.3. Image Processing and Data Augmentation

We processed the photographs using ImageJ 1.50 open-source image processing software [41]. Image sizes were $256 \times 256$ pixels. The following procedure was used for data augmentation of the training and test data. Data augmentation is a method for efficiently increasing the amount of learning data in deep learning. While deep learning does not require the definition of features, it requires the preparation of large volumes of training data. Because this is laborious, several methods have been proposed for efficient deep learning with less data. Typical methods include data augmentation and transfer learning [19,24,42].

First, we obtained three leaf images from each photograph by slightly varying the cropping window (Process 1 in Figure 2). Next, we rotated each cropped image clockwise by 90, 180 or 270 degrees (Process 2 in Figure 2). These procedures yielded 12 images per photograph and 3600 images per species; the final dataset comprised 72,000 images.

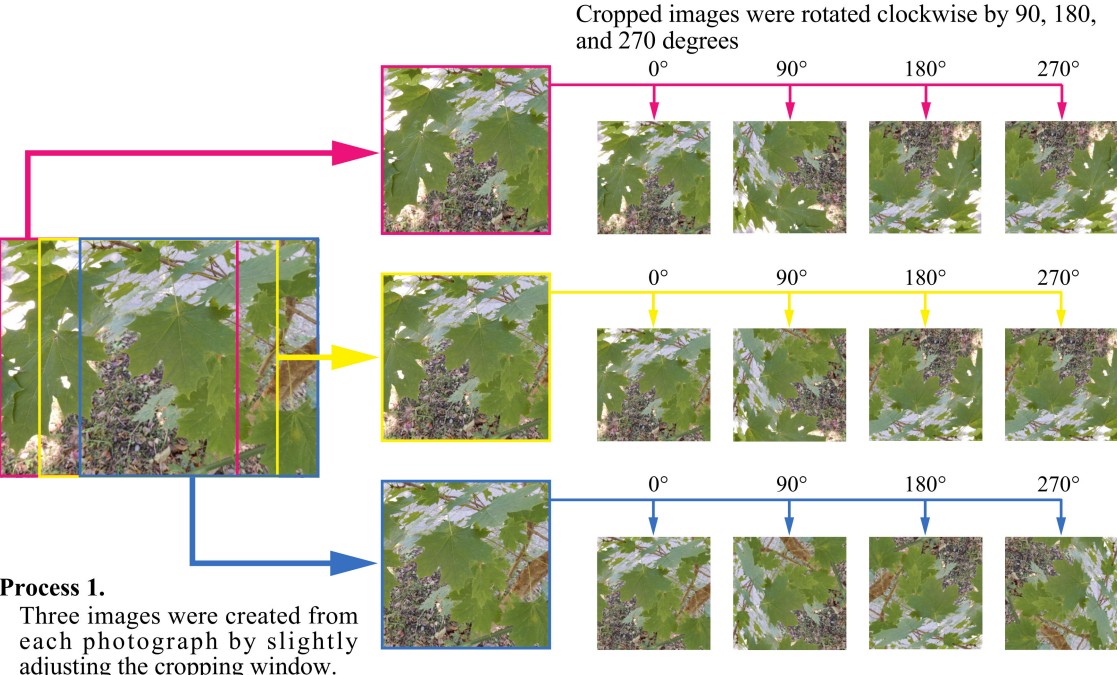

**Process 2.**
Cropped images were rotated clockwise by 90, 180, and 270 degrees

**Process 1.**
Three images were created from each photograph by slightly adjusting the cropping window.

**Figure 2.** Methods for creating leaf images from photographs.

### 2.4. CNN Algorithms

Of the available CNN algorithms, we chose AlexNet and GoogLeNet, which are representative algorithms from previous studies [5,6]. AlexNet, which won the ILSVRC 2012 competition, is a deep learning model that consists of five convolutional layers, three max-pooling layers, and three fully connected layers; it thus comprises 11 layers [43]. GoogLeNet, which won the ILSVRC 2014 competition, consists of 22 layers of stacked modules with nine inception modules that include multiple convolution and pooling layers [44]. The characteristics of GoogLeNet are similar to the characteristics of the network-in-network algorithm [45], which (a) incorporates a micro-network with fully connected feature maps, instead of the activation function; (b) has a network structure that exhibits depth and width in both the vertical and horizontal dimensions; and (c) applies the rectified linear unit activation function to all convolutional layers. Both algorithms use a dropout regularization method to suppress overfitting in fully connected layers.

### 2.5. Learning Environment and Models for the CNNs

The learning environment for both CNNs was a computer with a Linux operating system (Ubuntu 18.04 LTS), Intel Core i5–8400K central processing unit and NVIDIA GeForce RTX 2080 Super GPU [46]. We used CUDA 11.6 and cuDNN 8.3.2 to support the GPU with deep learning. We used convolutional architecture for fast feature embedding (Caffe, ver. 0.17.3) as the learning framework for the NVIDIA deep learning GPU training system (DIGITS) 6.1.1, which enables web-based learning. Caffe is a leading deep learning framework; it has hyper-parameter tuning, rapid execution speed and supports numerous operating systems [47].

### 2.6. Simulation Conditions and Performance Evaluation

We divided leaf images into 10 equal sets, each of which included all 20 species. We prepared two patterns of tree species identification methods, using data augmentation and no augmentation with each pattern. The first pattern identified the 20 tree species individually (hereafter referred to as "20-species classification"); the second pattern identified the 20 tree species in two groups, broadleaf and coniferous trees (hereafter referred to as "two-group classification"). The results from a previous study [5] suggest that deep learning avoids some of the challenges associated with human classification; these two methods were implemented to confirm this suspicion. Specifically, when a human misclassifies a tree species, the misclassification typically involves a tree species with similar shape features. For example, if a tree has simple leaves with smooth margins, it is likely to be misclassified as another species that also has simple leaves and smooth margins. However, deep learning has not shown such trends [5]. Misclassification might occur because the leaf image used for deep learning is not a scanned image that consists of a single leaf (used for tree species identification by machine learning using shape features)—it is a photograph acquired in the field, in which multiple leaves are reflected. Thus, misclassification is not limited to features that arise when deep learning is used. The application of deep learning to many tree species in the future will enable species identification, but there will be limitations in that coniferous trees may be mistakenly classified as broadleaf (or broadleaf trees may be mistakenly classified as coniferous). Furthermore, rather than classifying all cases at the species level in a single instance, it may be possible to classify the trees as broadleaf or coniferous in advance, then identify the species within each group. If this method demonstrated high accuracy, it would add to the repertoire of effective methods for tree species identification models with deep learning.

Here, we devised four learning models. The datasets for all models consisted of nine sets of training data and one set of test data. Learning Model 1 (LM-1) classified 20 tree species individually without data augmentation. Learning Model 2 (LM-2) classified 20 tree species individually with data augmentation. Learning Model 3 (LM-3) classified two groups (broadleaf and coniferous trees) without data augmentation. Finally, Learning Model 4 (LM-4) classified two groups (broadleaf and coniferous trees) with data augmen-

tation. We conducted 10 model iterations without duplication. LM-2 and LM-4 were prepared such that the 12 images enhanced by data augmentation were not divided into separate datasets (training and test); thus, the training and test datasets were independent. To evaluate the performance of the proposed models, we calculated the F-score, which was used for the classification accuracy of individual tree species (20 tree species for LM-1 and LM-2; two groups for LM-3 and LM-4) and the MCC, which was used for overall classification accuracy [48–50]. In a classification model that uses contingency tables, both true positive (*TP*) and true negative (*TN*) are correct classifications according to the classifier; TP is the positive example, while TN is the negative example, for each piece of training data. A false positive (*FP*) occurs when the outcome is incorrectly predicted to be "yes" (positive), although it is actually "no" (negative). A false negative (*FN*) occurs when the outcome is incorrectly predicted to be negative, although it is actually positive. The number of cases predicted to be positive that are actually positive refers to precision, while the number of cases that were actually correct and predicted to be positive is referred to as recall. Both are expressed by Equations (1) and (2), respectively.

$$Precision = \frac{TP}{TP + FN} \tag{1}$$

$$Recall = \frac{TP}{TP + FP} \tag{2}$$

Both equations are indices to evaluate classification accuracy, but they involve trade-offs [49,50]. In this study, the *F-score* (Equation (3)), which is the harmonic mean of the precision and recall, was used as an overall indicator.

$$F - score = \frac{2 \times Precision \times Recall}{Precision + Recall} \tag{3}$$

The *MCC* is an indicator that shows whether the classification is unbiased. It ranges from $-1$ to 1 [48]. The MCC was calculated using Equation (4) for LM-1 and LM-2 (because they are multi-class classifiers [51]) and Equation (5) for LM-3 and LM-4 (because they are binary classifiers). The MCC of Equation (4) can be defined by a confusion matrix $C$ for $K$ classes [51].

$$MCC = \frac{c \times s - \sum_k^K p_k \times t_k}{\sqrt{\left(s^2 - \sum_k^K p_k^2\right) \times \left(s^2 - \sum_k^K t_k^2\right)}} \tag{4}$$

where $t_k = \sum_i^K C_{ik}$ is the number of times class $k$ actually occurs in the confusion matrix $C$ of class $K$; $p_k = \sum_i^K C_{ki}$ is the number of times class $k$ was predicted in the confusion matrix $C$ of class $K$; $c = \sum_k^K C_{kk}$ is the total number of samples correctly predicted; and $s = \sum_i^K \sum_j^K C_{ij}$ is the total number of samples [51].

$$MCC = \frac{TP \times TN - FP \times FN}{\sqrt{(TP + FP)(TP + FN)(TN + FP)(TN + FN)}} \tag{5}$$

We conducted 50, 100 and 200 epochs for all learning models.

## 3. Results

### 3.1. Classification Accuracy of Tree Species Identification for Test Data When Simultaneously Identifying Broadleaf and Coniferous Trees

Table 1 shows F-scores that represent the classification accuracies of tree species for test data when broadleaf and coniferous tree species were classified simultaneously. The values in the table are the means of 10 simulation results.

**Table 1.** Classification accuracy of tree species identification for test data when simultaneously classifying broadleaf and coniferous trees.

| Categories | | Tree Species | No Data Augmentation | | | | | | Data Augmentation | | | | | |
|---|---|---|---|---|---|---|---|---|---|---|---|---|---|---|
| | | | GoogLeNet | | | AlexNet | | | GoogLeNet | | | AlexNet | | |
| | | | 50 [1] | 100 [1] | 200 [1] | 50 [1] | 100 [1] | 200 [1] | 50 [1] | 100 [1] | 200 [1] | 50[1] | 100 [1] | 200 [1] |
| Broadleaf trees | Smooth margins [2] | C.c. | 0.912 | 0.950 | 0.948 | 0.899 | 0.914 | 0.923 | 0.966 | 0.972 | 0.975 | 0.949 | 0.956 | 0.960 |
| | | C.j.N | 0.881 | 0.898 | 0.906 | 0.802 | 0.846 | 0.839 | 0.959 | 0.962 | 0.969 | 0.916 | 0.925 | 0.935 |
| | | M.t. | 0.846 | 0.917 | 0.914 | 0.810 | 0.859 | 0.850 | 0.953 | 0.964 | 0.974 | 0.942 | 0.944 | 0.956 |
| | | S.s. | 0.870 | 0.916 | 0.929 | 0.777 | 0.845 | 0.844 | 0.961 | 0.963 | 0.977 | 0.931 | 0.943 | 0.947 |
| | Toothed margins [3] | C.j.Z | 0.943 | 0.946 | 0.962 | 0.913 | 0.939 | 0.940 | 0.974 | 0.975 | 0.981 | 0.965 | 0.966 | 0.972 |
| | | L.g. | 0.788 | 0.847 | 0.872 | 0.740 | 0.761 | 0.766 | 0.924 | 0.925 | 0.940 | 0.885 | 0.895 | 0.904 |
| | | Q.g. | 0.863 | 0.904 | 0.908 | 0.808 | 0.833 | 0.834 | 0.952 | 0.962 | 0.969 | 0.928 | 0.945 | 0.947 |
| | | Q.p. | 0.841 | 0.926 | 0.928 | 0.797 | 0.861 | 0.885 | 0.969 | 0.976 | 0.985 | 0.944 | 0.954 | 0.958 |
| | Lobed [4] | A.p. | 0.793 | 0.859 | 0.868 | 0.735 | 0.764 | 0.777 | 0.941 | 0.944 | 0.954 | 0.900 | 0.908 | 0.910 |
| | | A.p.ck | 0.826 | 0.908 | 0.918 | 0.821 | 0.849 | 0.856 | 0.965 | 0.966 | 0.977 | 0.940 | 0.959 | 0.953 |
| | | G.b. | 0.929 | 0.949 | 0.968 | 0.925 | 0.942 | 0.941 | 0.964 | 0.969 | 0.972 | 0.956 | 0.968 | 0.963 |
| | | L.f. | 0.774 | 0.846 | 0.840 | 0.687 | 0.743 | 0.763 | 0.923 | 0.927 | 0.945 | 0.899 | 0.905 | 0.902 |
| | | Min. | 0.774 | 0.846 | 0.840 | 0.687 | 0.743 | 0.763 | 0.923 | 0.925 | 0.940 | 0.885 | 0.895 | 0.902 |
| | | Max. | 0.943 | 0.950 | 0.968 | 0.925 | 0.942 | 0.941 | 0.974 | 0.976 | 0.985 | 0.965 | 0.968 | 0.972 |
| | | Ave. | 0.855 | 0.906 | 0.914 | 0.809 | 0.846 | 0.852 | 0.954 | 0.959 | 0.968 | 0.930 | 0.939 | 0.942 |
| Coniferous trees | | A.f. | 0.756 | 0.792 | 0.818 | 0.673 | 0.729 | 0.731 | 0.875 | 0.857 | 0.871 | 0.849 | 0.861 | 0.869 |
| | | C.f. | 0.805 | 0.812 | 0.847 | 0.765 | 0.770 | 0.776 | 0.857 | 0.867 | 0.873 | 0.846 | 0.843 | 0.833 |
| | | C.j.D | 0.768 | 0.829 | 0.861 | 0.745 | 0.750 | 0.758 | 0.920 | 0.925 | 0.930 | 0.909 | 0.903 | 0.904 |
| | | C.o. | 0.826 | 0.862 | 0.883 | 0.696 | 0.762 | 0.776 | 0.929 | 0.923 | 0.935 | 0.880 | 0.897 | 0.886 |
| | | M.g. | 0.872 | 0.912 | 0.917 | 0.863 | 0.881 | 0.874 | 0.929 | 0.918 | 0.934 | 0.907 | 0.911 | 0.914 |
| | | P.t. | 0.890 | 0.913 | 0.929 | 0.837 | 0.826 | 0.845 | 0.942 | 0.946 | 0.954 | 0.942 | 0.954 | 0.944 |
| | | T.c. | 0.751 | 0.801 | 0.812 | 0.619 | 0.678 | 0.697 | 0.842 | 0.851 | 0.883 | 0.812 | 0.796 | 0.823 |
| | | T.s. | 0.834 | 0.888 | 0.899 | 0.739 | 0.784 | 0.813 | 0.931 | 0.942 | 0.952 | 0.900 | 0.903 | 0.907 |
| | | Min. | 0.751 | 0.792 | 0.812 | 0.619 | 0.678 | 0.697 | 0.842 | 0.851 | 0.871 | 0.812 | 0.796 | 0.823 |
| | | Max. | 0.890 | 0.913 | 0.929 | 0.863 | 0.881 | 0.874 | 0.942 | 0.946 | 0.954 | 0.942 | 0.954 | 0.944 |
| | | Ave. | 0.813 | 0.851 | 0.871 | 0.742 | 0.772 | 0.784 | 0.903 | 0.904 | 0.917 | 0.880 | 0.884 | 0.885 |
| | Min. for overall | | 0.751 | 0.792 | 0.812 | 0.619 | 0.678 | 0.697 | 0.842 | 0.851 | 0.871 | 0.812 | 0.796 | 0.823 |
| | Max. for overall | | 0.943 | 0.950 | 0.968 | 0.925 | 0.942 | 0.941 | 0.974 | 0.976 | 0.985 | 0.965 | 0.968 | 0.972 |
| | Ave. for overall | | 0.838 | 0.884 | 0.896 | 0.782 | 0.817 | 0.824 | 0.934 | 0.937 | 0.947 | 0.910 | 0.917 | 0.919 |

[1] Epochs, [2] Simple leaves with smooth margins, [3] Simple leaves with toothed margins, [4] Lobed leaves.

Without data augmentation, the highest F-score was 0.968 for G.b. (please see abbreviations list for all species names used in this manuscript) in GoogLeNet after 200 epochs. The lowest F-score was 0.619 for T.c. in AlexNet after 50 epochs. The highest F-scores of broadleaf trees in GoogLeNet were obtained for 0.943 for C.j.Z. after 50 epochs, 0950 for C.c. after 100 epochs and 0.968 for G.b. after 200 epochs; thus, the results differed according to the number of epochs. In AlexNet, G.b. had the highest F-scores for all epochs (0.925, 0.942 and 0.941 in order of decreasing epoch). The lowest F-scores from the broadleaf tree category were L.f. for all epochs in both GoogLeNet and AlexNet (0.774, 0.846 and 0.840 in order of decreasing epochs for GoogLeNet; 0.687, 0.743 and 0.763 for AlexNet). The highest coniferous tree F-scores in GoogLeNet were P.t. for all epochs (0.890, 0.913 and 0.929 in order of decreasing epoch); the highest coniferous F-scores in AlexNet were M.g. for all epochs (0.863, 0.881 and 0.874 in order of decreasing epoch). The lowest coniferous tree F-scores in GoogLeNet were 0.751 for T.c. after 50 epochs, 0.792 for A.f. after 100 epochs and 0.812 for T.c. after 200 epochs; the lowest coniferous tree F-scores in AlexNet were T.c. for all epochs (0.619, 0.678 and 0.697 in order of decreasing epoch).

Comparison of overall F-scores by epoch revealed that minimum, maximum and mean values were higher for broadleaf trees across all epochs for both GoogLeNet and AlexNet. Similar to differences among epochs, the F-score tended to increase as epochs increased for most learning patterns, although some patterns showed slightly lower F-scores (C.c., M.t. and L.f. in GoogLeNet after 200 epochs; P.t. in AlexNet after 100 epochs; and C.j.N., M.t., S.s., G.b. and M.g. in AlexNet after 200 epochs). Among the learning algorithms used, GoogLeNet had a higher F-score for all training patterns.

With data augmentation, the highest F-score was 0.985 for Q.p. in GoogLeNet after 200 epochs, while the lowest was 0.796 for T.c. in AlexNet after 100 epochs. With regard to epoch, the highest broadleaf tree F-scores in GoogLeNet were 0.974 for C.j.Z. after 50 epochs, 0.976 for Q.p. after 100 epochs and 0.985 for Q.p. after 200 epochs; the highest broadleaf tree F-scores in AlexNet were 0.965 for C.j.Z. after 50 epochs, 0.968 for G.b. after 100 epochs and 0.972 for C.j.Z. after 200 epochs. The lowest broadleaf tree F-scores in GoogLeNet

were 0.923 for L.f. after 50 epochs, 0.925 for L.g. after 100 epochs and 0.940 for L.g. after 200 epochs; the lowest broadleaf tree F-scores in AlexNet were 0.885 for L.g. after 50 epochs, 0.895 for L.g. after 100 epochs and 0.902 for L.f. after 200 epochs. The highest coniferous tree F-scores in GoogLeNet were P.t. for all epochs (0.942, 0.946 and 0.954 in order of decreasing epoch); the highest coniferous tree F-scores in AlexNet were P.t. for all epochs (0.942, 0.954 and 0.944 in order of decreasing epoch). The lowest coniferous tree F-scores in GoogLeNet were 0.842 for T.c. after 50 epochs, 0.851 for R.c. after 100 epochs and 0.871 for A.f. after 200 epochs; the lowest coniferous tree F-scores in AlexNet were T.c. for all epochs (0.812, 0.796 and 0.823 in order of decreasing epoch).

Comparison of overall F-scores by epoch revealed that minimum, maximum and mean values were higher for broadleaf trees across all epochs for both GoogLeNet and AlexNet, similar to the findings without data augmentation. Similar to differences among epochs, the F-score tended to increase as epochs increased for most of the learning patterns, but some of the patterns showed slightly lower F-scores, similar to the findings without data augmentation (A.f., C.o. and M.g. in GoogLeNet after 100 epochs; C.f., C.j.D. and T.c in AlexNet after 100 epochs; and A.p.ck, G.b., L.f., Co., C.f. and P.t. in both GoogLeNet and AlexNet after 200 epochs). Among the learning algorithms used, GoogLeNet had a higher F-score for many learning patterns; in a few cases, AlexNet had a higher score (A.f. and P.t. in AlexNet after 100 epochs).

### 3.2. Classification Accuracy of Tree Species Identification for Test Data When Either Broadleaf or Coniferous Trees Were Used

Table 2 shows F-scores that represent the classification accuracies of tree species for test data when classifying between groups of either broadleaf or coniferous trees. The values in the table are the means of 10 simulation results.

**Table 2.** F-scores as the classification accuracy of tree species for test data when classifying between groups of either broadleaf or coniferous trees.

| Groups | Data Augmentation | GoogLeNet | | | AlexNet | | |
|---|---|---|---|---|---|---|---|
| | | **50** [1] | **100** [1] | **200** [1] | **50** [1] | **100** [1] | **200** [1] |
| Broadleaf | - | 0.9965 | 0.9967 | 0.9967 | 0.9935 | 0.9953 | 0.9960 |
| | ○ [2] | 0.9981 | 0.9988 | 0.9982 | 0.9978 | 0.9983 | 0.9981 |
| Coniferous | - | 0.9948 | 0.9950 | 0.9950 | 0.9902 | 0.9929 | 0.9940 |
| | ○ [2] | 0.9972 | 0.9982 | 0.9974 | 0.9967 | 0.9974 | 0.9972 |

[1] Epochs, [2] Open circles indicate with data augmentation.

The highest F-score was 0.9988 for broadleaf trees with data augmentation in GoogLeNet after 100 epochs. The lowest F-score was 0.9902 for coniferous trees without data augmentation in AlexNet after 50 epochs. Overall, F-scores were very high for 20-species classification. F-scores were higher with data augmentation for all epochs. Among the various epochs, F-scores tended to increase as epochs increased, but the difference was very small. GoogLeNet after 200 epochs with data augmentation resulted in a slightly lower F-score than after 100 epochs for both broadleaf and coniferous trees. Comparison of the highest F-score for each tree species revealed almost identical results for broadleaf and coniferous trees: 0.9988 and 0.9982, respectively (both with data augmentation in GoogLeNet after 100 epochs). The number of errors was 51 of 42,000 images for broadleaf trees and 53 of 28,800 images for coniferous trees; these results indicated very high classification accuracy for group.

### 3.3. Classification Accuracy of Tree Species Identification According to Learning Method

Figure 3 shows the classification accuracy of tree species identification according to learning method for test data, on the basis of MCC. In the legend for Figure 3, "20 species individually" indicates LM-1 (20-species classification); "only broadleaf" indicates the

MCC, which was calculated using results from Ref. [5]; "only coniferous" indicates the MCC, which was calculated using results from Ref. [6]; and "two groups" indicates LM-3 (two-group classification). Figure 3a shows GoogLeNet and Figure 3b shows AlexNet; solid lines indicate data augmentation and dashed lines indicate no data augmentation.

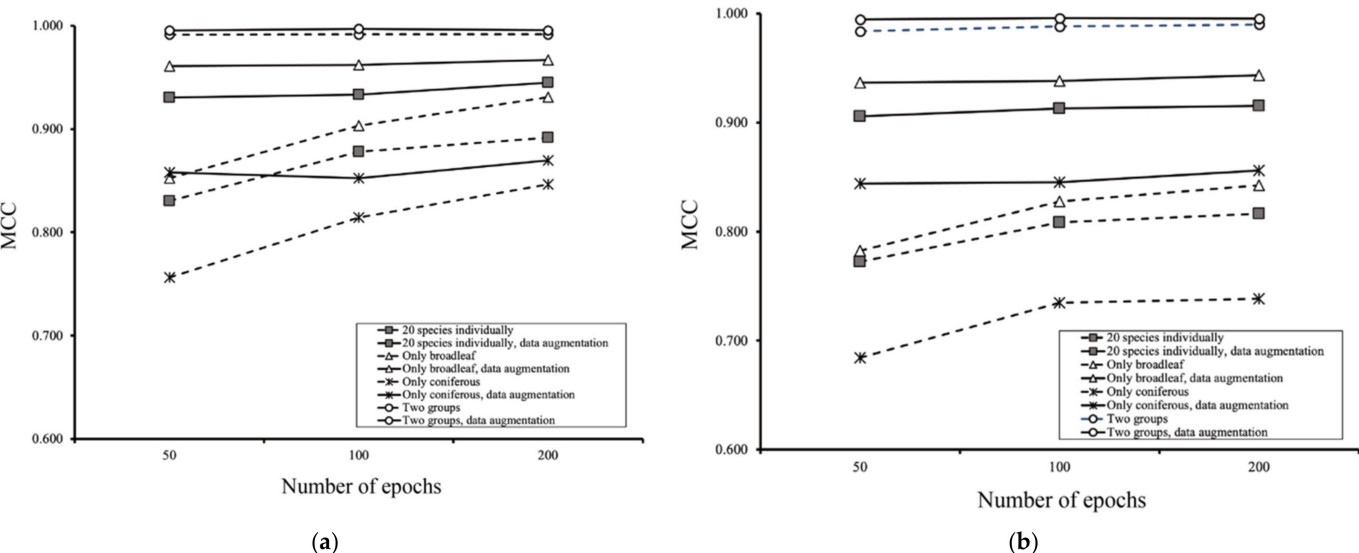

(**a**)                                                                                                                       (**b**)

**Figure 3.** Classification accuracy of tree species identification for test data, based on MCC. (**a**) GoogLeNet; (**b**) AlexNet.

In Figure 3a (GoogLeNet), the highest MCC was 0.997 for "two groups, data augmentation" after 100 epochs. MCCs for "two groups, data augmentation" were high for all epochs. The lowest MCC was 0.756 for "only coniferous" after 50 epochs. Comparison of the results with and without data augmentation revealed that classification accuracy tended to improve as epochs increased in most cases without data augmentation. The two-group classifications ("two groups" and "two groups, data augmentation" in Figure 3a) showed a very high classification accuracy of 0.995 with data augmentation and 0.991 without data augmentation after 50 epochs; thus, there was almost no improvement in classification accuracy as epochs increased. With data augmentation, there was no improvement in classification accuracy as epochs increased; in some cases, there was a slight decrease in classification accuracy as epochs increased ("only coniferous, data augmentation" after 100 epochs and "two groups, data augmentation" after 200 epochs). In Figure 3b (AlexNet), the highest MCC was 0.996 for "two groups, data augmentation" after 100 epochs. MCCs for "two groups, data augmentation" were high for all epochs, similar to the GoogLeNet results. The lowest MCC was 0.684 for "only coniferous" after 50 epochs. The AlexNet results with and without data augmentation were similar to the GoogLeNet results. The classification accuracy in AlexNet was almost identical to the classification accuracy in GoogLeNet for "two groups, data augmentation"; however, the accuracy of the other groups tended to be lower overall, especially concerning MCCs without data augmentation (which were lower with the exception of "two groups").

*3.4. Comparison between Simultaneous and Individual Identification*

As indicated in Section 3.3, the two-group classification had the highest accuracy, followed by only broadleaf trees, 20-species classification individually and coniferous trees only. Therefore, the accuracy of tree species identification was higher for broadleaf trees when only broadleaf trees were used; it was higher for coniferous trees when both broadleaf and coniferous trees were identified simultaneously. However, because the above MCC is calculated for the entire classification, the MCC when both broadleaf and coniferous trees were classified simultaneously may be the result of averaging the classification accuracies of broadleaf trees only and coniferous trees only. Therefore, from the cases in which broadleaf

and coniferous trees were classified simultaneously, the respective results for broadleaf and coniferous trees were extracted and compared with the results from broadleaf only classification [5] and coniferous only classification [6], respectively (Table 3).

**Table 3.** Comparison between simultaneous identification of broadleaf and coniferous trees and individual identification of broadleaf and coniferous trees.

| Pattern | Data Augmentation | GoogLeNet | | | AlexNet | | |
|---|---|---|---|---|---|---|---|
| | | 50 [1] | 100 [1] | 200 [1] | 50 [1] | 100 [1] | 200 [1] |
| SB [2] | - | −31 | −24 | −84 | 35 | 20 | −6 |
| | ○ [4] | −399 | −275 | −76 | −470 | −150 | −212 |
| SC [3] | - | 65 | 39 | 16 | 49 | 14 | 30 |
| | ○ [4] | 772 | 945 | 900 | 462 | 536 | 294 |

[1] Epochs, [2] Total value of TP according to tree group when classified simultaneously minus the TP value of broadleaf trees only, [3] Total value of TP according to tree group when classified simultaneously minus the TP value of coniferous trees only, [4] Open circles indicate with data augmentation.

Notably, the number of misclassified tree species differed for each training model; comparisons using MCCs and F-scores were thus inappropriate. We compared the total TPs, which represent the correct classification of the positive examples, according to tree group. The values in Table 3 show the total TP value according to tree group when classified simultaneously minus the TP value of broadleaf or coniferous only, respectively. Positive (+) values indicate a higher accuracy in the classification of broadleaf and coniferous trees simultaneously, while negative (−) values indicate a higher accuracy when only broadleaf trees or only coniferous trees are classified. For broadleaf trees in GoogLeNet, all epochs with and without data augmentation showed negative values. Therefore, we presumed that the accuracy of tree species identification would be higher when learning only broadleaf trees than when learning the classification of broadleaf and coniferous trees simultaneously. Without data augmentation, the classification accuracy was lower after 100 epochs than after 50 epochs, but the overall trend was that the classification accuracy increased as epochs increased. With data augmentation, the classification accuracy tended to decrease as epochs increased. For broadleaf trees in AlexNet, the results after 50 and 100 epochs without data augmentation indicated increased classification accuracy when trees were classified simultaneously. However, classification accuracy tended to increase as epochs increased when broadleaf trees only were used for training. For coniferous trees in both GoogLeNet and AlexNet, all epochs with and without data augmentation showed positive values. Therefore, we presumed that the accuracy of tree species identification would be higher when learning the classification of broadleaf and coniferous trees simultaneously than when learning coniferous trees only. GoogLeNet showed a trend toward higher classification accuracy for coniferous trees only without data augmentation and for data augmentation with simultaneous classification as epochs increased.

### 3.5. Misclassification Patterns According to Tree Species

Figures 4 and 5 show misclassification patterns according to tree species. The figures comprise heatmaps that were created based on the classification results obtained from the contingency table, using the seaborn module in Python. Figure values show classification according to each training model; they represent the sum of the results of 10 simulations. Based on the results in Table 3, the results with the lowest and highest MCCs are shown in Figures 4 and 5, respectively. Training class represents each tree species in the training cases; test class represents where the training model classified each tree species (i.e., how the training model classified each tree species used in the training cases). The figure shows that the correct classification was made only when the training class and the test class had the same tree species. To clarify the misclassification, broadleaf and coniferous trees are indicated along with three groups of broadleaf trees based on leaf shape, which are divided and clearly indicated using solid lines.

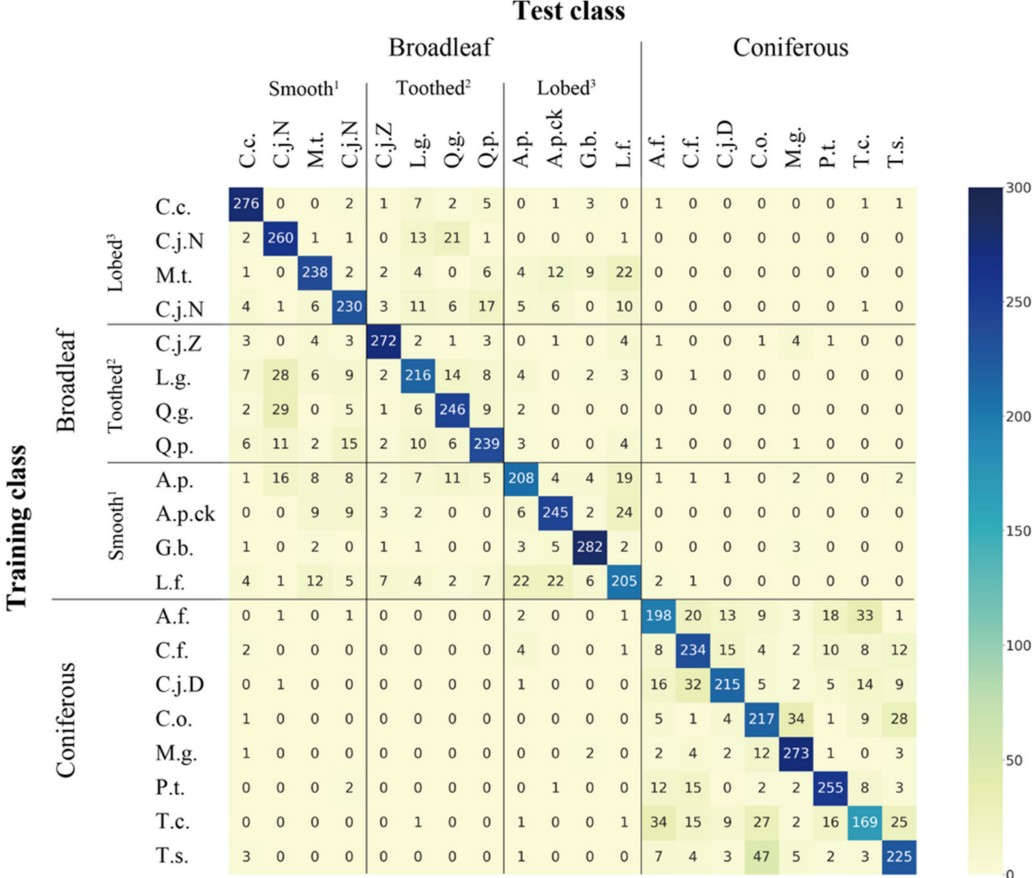

**Figure 4.** Misclassification patterns according to tree species in AlexNet after 50 epochs without data augmentation. [1] Simple leaves with smooth margins, [2] Simple leaves with toothed margins, [3] Lobed leaves.

In Figure 4, the highest TP was G.b. and the lowest TP was T.c. Overall, there were few misclassifications of broadleaf trees as coniferous trees or coniferous trees as broadleaf trees. Both broadleaf and coniferous trees were frequently misclassified within each category; broadleaf trees were misclassified within three categories. The four species of broadleaf trees that were not misidentified as coniferous trees were M.t., C.j.N., Q.g. and A.p.ck. There were no cases in which coniferous trees were misidentified as broadleaf trees among any of the tree species. Except for T.c., seven coniferous trees had no misclassification of broadleaf trees as a category of simple leaves with toothed margins. Comparison of the four categories (three broadleaf categories and one coniferous) revealed that the categories with the highest total number of misclassifications for each category were the misclassification of simple leaves with smooth margins as simple leaves with toothed margins (99 total misclassifications), simple leaves with toothed margins as simple leaves with smooth margins (130), lobed leaves as coniferous leaves (119) and coniferous leaves as lobed leaves (568).

In Figure 5, the highest TP was S.s. and the lowest TP was A.f. Similar to Figure 4, there were few cases of misclassification of broadleaf trees as coniferous trees or coniferous trees as broadleaf trees. Both broadleaf and coniferous trees were frequently misclassified within each category; broadleaf trees were misclassified within three categories. The five species for which broadleaf trees were not misidentified as coniferous trees were M.t., C.j.N., Q.g., Q.p. and A.p.ck. The three species for which coniferous trees were not misidentified as broadleaf trees were C.o., M.g. and T.c. There were no misclassifications for simple leaves with toothed margins and lobed leaves on A.f. and P.t. or for simple leaves with smooth leaves and lobed leaves on C.f. Comparison of the four categories of broadleaf and three categories of coniferous trees revealed that the categories with the highest total

number of misclassifications for each category were the misclassification of simple leaves with smooth margins as simple leaves with toothed margins (186), simple leaves with toothed margins as simple leaves with toothed margins (168), lobed leaves as simple leaves with smooth margins (245) and coniferous leaves as lobed leaves (2339).

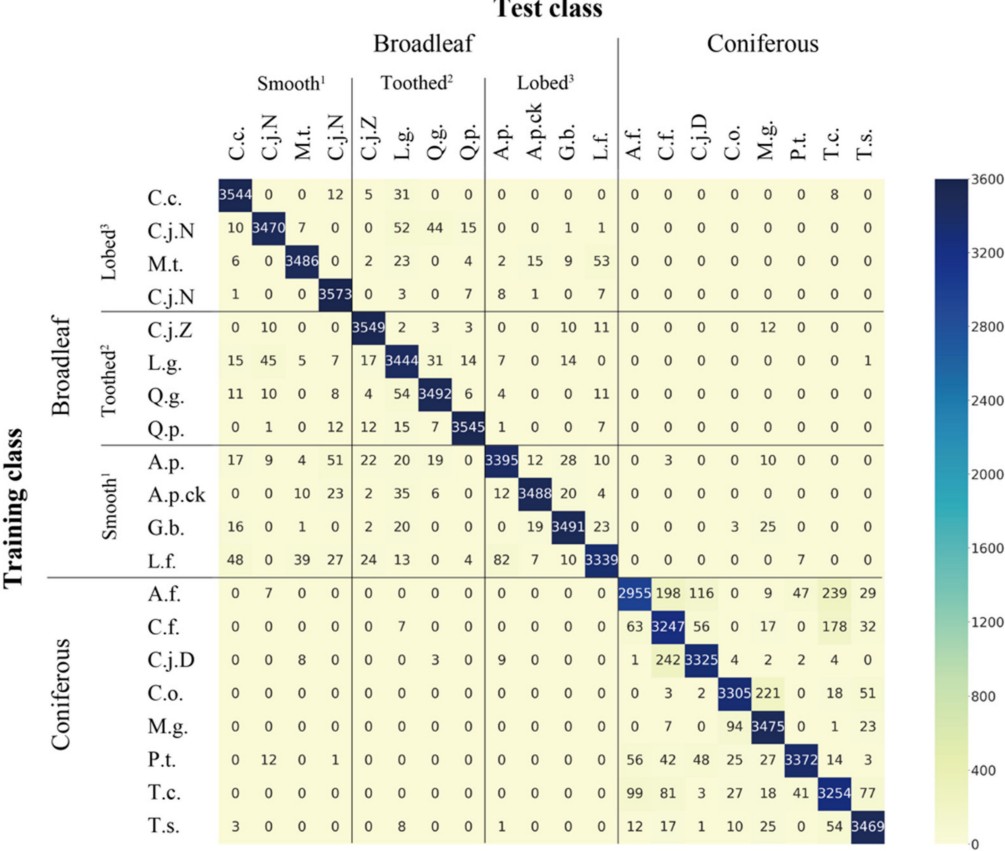

**Figure 5.** Misclassification patterns according to tree species in GoogLeNet after 200 epochs with data augmentation. [1] Simple leaves with smooth margins, [2] Simple leaves with toothed margins, [3] Lobed leaves.

## 4. Discussion

In this study, when 12 broadleaf and eight coniferous trees were classified simultaneously, the highest MCC was 0.8914 without data augmentation and 0.9447 with data augmentation. While the model with the highest MCC for only 12 broadleaf trees had a classification accuracy of 0.9309 without data augmentation and 0.9670 with data augmentation [5], only eight coniferous trees had a classification accuracy of 0.8465 without data augmentation and 0.8695 with data augmentation [6]. Considering the results in Table 3, the classification accuracy was higher for broadleaf trees when the model was trained using broadleaf only, whereas the classification accuracy of coniferous trees was higher when the model was trained using both tree categories simultaneously than when it was trained with coniferous trees only. These findings should be verified in additional studies because of the difference in the number of broadleaf and coniferous trees; moreover, the leaf images were acquired from only a few trees of each species.

With respect to epochs, many models showed a trend toward improved classification accuracy as epochs increased. Especially without data augmentation, there was substantial improvement in classification accuracy as epochs increased. While high classification accuracy was achieved after 50 epochs (the number of epochs used as the fewest number of training sessions), there was minimal improvement in classification accuracy as epochs increased thereafter; there were a few cases in which classification accuracy slightly decreased. In most types of machine learning, including deep learning, the use of excessive

learning iterations (i.e., epochs in deep learning) can lead to low classification accuracy of test data because the training data are overfitted to the training cases; this is known as "overfitting" [52]. In the present study, overfitting presumably occurred in the two-group classification as epochs increased because these models achieved sufficient classification accuracy with a small number of epochs.

With respect to learning algorithms, GoogLeNet tended to have higher classification accuracy than AlexNet for most of the models. Because of the similar results in tree species identification in 2015 PlantCLEF and Ghazi et al. [17] mentioned above, GoogLeNet is regarded as one of the most effective deep learning algorithms with high classification accuracy. A possible reason for GoogLeNet to exhibit higher classification accuracy is that GoogLeNet has more layers than AlexNet. Deep learning tends to improve image recognition performance as the number of layers increases. However, several challenges have been identified in the application of deep learning. Constraints on input data (i.e., image type and image size) vary among algorithms; moreover, algorithms with many layers require extensive computation as the number of layers increases, which leads to long training times or the need for high performance hardware. Therefore, it is preferable to consider the optimal type of learning model for different problem sets, rather than simply using a model with many layers.

With respect to data augmentation, the classification accuracy tended to improve with data augmentation when many objects were identified. There was also sufficient classification accuracy without data augmentation when identifying both broadleaf and coniferous trees (two-group classification). This finding is presumably because a large amount of training data was used for two-group classification: 3600 images for broadleaf trees and 2400 images for coniferous trees. If high classification accuracy was achieved without data augmentation, data augmentation led to a slight decrease in classification accuracy. In general, deep learning requires the preparation of a large amount of training data for analysis, but there is no specific threshold for analysis target data. Based on our results, determination of the need for data augmentation should be based on the classification target and the amount of data. If classification accuracy is high despite a small amount of training data, data augmentation is unnecessary; in such cases, an unnecessarily large number of epochs can decrease classification accuracy because of overfitting.

With respect to misclassification patterns, broadleaf trees were often misclassified as the incorrect broadleaf species and coniferous trees were often misclassified as the incorrect coniferous species. However, because identification is not 100% correct, there were a few cases where broadleaf trees were misclassified as coniferous trees or coniferous trees were misclassified as broadleaf trees; such misclassification is unlikely to occur with human identification. Although detailed verification was not conducted in this study because it would been extremely complicated, image recognition using deep learning can output classification results as rankings, as in the mean reciprocal rank method described earlier, and in the DIGITS method used in this study, where the classification results are displayed as probabilities with the top five rankings. Therefore, when constructing a practical tree species identification system, it is preferable to use a method that can present several candidates instead of a single tree species. In addition, misidentifications (e.g., confusion between broadleaf and coniferous trees) can be prevented if candidate tree species are expressed as probabilities or if images of tree species and textual information about tree species are presented simultaneously.

Based on the results of this study, we propose the following learning method for constructing a tree species identification system. First, distinguish between two groups: broadleaf and coniferous trees. Then, perform classification of broadleaf trees with a learning model for broadleaf trees only; this will provide higher classification accuracy. For coniferous trees, use a classification model for broadleaf and coniferous trees simultaneously; this will provide higher classification accuracy. Notably, we found distinct trends in species identification between broadleaf and coniferous trees, but the reasons for these differences are unclear. The number of tree species used in this study was small; we expect

that the trend would have differed if more tree species had been used. To examine the accuracy of tree species identification using deep learning, we investigated deep learning algorithms, learning patterns, epochs and the amounts of training and test data with and without data augmentation. To improve the accuracy of tree species classification, there is a need to improve the classification accuracy of coniferous trees in particular. Considering that the species classification accuracy is lower for coniferous trees than for broadleaf trees (although the number of coniferous species is smaller than the number of broadleaf tree species), learning conditions should be considered along with other conditions (e.g., learning using leaf images, as well as images of whole trees and stems). Furthermore, we are exploring the construction of a system that can perform tree species identification by implementing the results of machine and deep learning identification that have been conducted on mobile terminals. To achieve tree species identification using mobile terminals, machine and deep learning must be implemented on the mobile terminals. We are conducting tree species identification using deep learning for mobile terminals such as MobileNetV3 [53] and GhostNet [54], which are typical deep learning algorithms. Alternatively, rather than direct implementation on a mobile device, CoreML for iOS [37] and ML Kit for Android [55] have been proposed for training on non-mobile devices; these softwares allow the results to subsequently be applied to a mobile device. Empirical studies using these methods will be important in the future.

## 5. Conclusions and Future Work

We applied a deep learning technique to identify tree species based on outdoor photographs of leaves from 12 broadleaf and eight conifer species. We evaluated the accuracy of our results by classifying broadleaf and conifer trees simultaneously. Outdoor tree leaf photographs typically include multiple leaves, rather than single leaves. Although single-leaf images or those adjusted for brightness or saturation are associated with high tree species classification accuracy [30–34], our results show that multiple-leaf images acquired in the field could be used to classify tree species with high accuracy. Variation in classification accuracy and misclassification was observed for both broadleaf and conifer trees. Further study is required to verify the results of this study using a larger number of tree species. In this study, we did not examine the effects of the conditions under which photographs were taken. It is possible that different backgrounds for the same tree species in various locations may have affected classification accuracy. For example, a previous study found that leaf segmentation or overlapping strongly influenced the classification of broadleaf tree images, particularly in images with complex backgrounds, necessitating labeling for segmentation [32]. We conclude that labeling should also be considered for coniferous trees, although it would increase the labor required for image processing.

**Author Contributions:** Conceptualization, Y.M.; methodology, Y.M.; software, Y.M.; validation, Y.M., Y.K. and S.N.; formal analysis, Y.M., Y.K. and S.N.; investigation, Y.K. and S.N.; resources, Y.K. and S.N.; data curation, Y.M., Y.K. and S.N.; writing—original draft preparation, Y.M., Y.K. and S.N.; writing—review and editing, Y.M.; visualization, Y.M.; supervision, Y.M.; project administration, Y.M. All authors have read and agreed to the published version of the manuscript.

**Funding:** This research received no external funding.

**Data Availability Statement:** Not applicable.

**Acknowledgments:** We are grateful to the members of the Laboratory of Watershed Information Science and the Laboratory of Erosion Control, Kyoto Prefectural University, for their cooperation in conducting this study.

**Conflicts of Interest:** The authors declare no conflict of interest.

## Abbreviations

The following abbreviations are used in this manuscript:

| | |
|---|---|
| CNN | convolutional neural network |
| MCC | Matthews correlation coefficient |
| TP | true positive |
| TN | true negative |
| FP | false positive |
| FN | false negative |
| C.c. | *Cinnamomun camphora* (Linn.) Sieb. |
| C.j.N | *Cinnamomum japonicum* Sieb. ex Nakai |
| M.t. | *Maclura tricuspidata* (syn. Cudrania tricuspidata) |
| S.s. | *Sapium sebiferum* (Linn.) Roxb. |
| C.j.Z | *Cercidiphyllum japonicum* Sieb. et Zucc. |
| L.g. | *Lithocarpus glaber* Thunb. |
| Q.g. | *Quercus glauca* Thunb. |
| Q.p. | *Quercus philyraeoides* A. Gray |
| A.p. | *Acer palmatum* Thunb. |
| A.p.ck | *Acer platanoides* cv. crimson king |
| G.b. | *Ginkgo biloba* Linn. |
| L.f. | *Liquidambar formosana Hance.* |
| A.f. | *Abies firma* |
| C.f. | *Calocedrus formosana* |
| C.j.D | *Cryptomeria japonica* D. Don |
| C.o. | *Chamaecyparis obtusa* |
| M.g. | *Metasequoia glyptostroboides* |
| P.t. | *Pinus thunbergii* |
| T.c. | *Taxus cuspidata* |
| T.s. | *Tsuga sieboldii* |

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
