# Peer review of "Verification of a Deep Learning-Based Tree Species Identification Model Using Images of Broadleaf and Coniferous Tree Leaves"

_forests, doi:10.3390/f13060943_

Round 1

Reviewer 1 Report

Very valuable paper, however my concern is related to following issues:

- how image background features are affecting to results, thus non-leaf areas?? If single species is taken with similar conditions, and lead is not extracted separately, it effects to results.

- how imaging conditions affect to results, thus imaging angles will certainly affect, but how much??

- The used methods are described rather general way.

- The verification concept and use of leave-one-out-type of design is definetly needed in this approach, but I was not sure if that was used. Please clarify evaluation methods.

Reviewer 2 Report

Verification of a Deep Learning-based Tree Species Identification Model using Images of Broadleaf and Coniferous Tree Leaves was submitted to review. Authors already published different papers in the field. This paper is correctly written and organized.

The objective of this study was to verify the accuracy of tree species identification by using deep learning with broadleaf and coniferous leaf images. Authors have got a very good accuracy by using two popular algorithms. Acquired images are preprocessed and augmented in order to have appropriate dataset for training and testing. Finally, to examine the accuracy of tree species identification using deep learning, authors investigated deep learning algorithms, learning patterns, epochs, and the amounts of training and test data with and without data augmentation.

I have some considerations for this work and the authors need to explain and modify the paper. Some of these are:

I would like authors to emphasize contribution of their research

Authors are aware that the number of tree species used in this study is small. Is there a possibility to improve it?

Please try to find newest literature on the topic, there are 43 references but they are in average 5-10 years old.

Why different cameras are used for broadleaf and coniferous images acquisition? Are the authors sure this fact doesn’t have impact on the results. This needs to be discussed.

Line 97: We used 12 broadleaf and eight coniferous trees – 12 or twelve (eight or 8) – would prefer same style

Finally, please try to give some conclusions and future work proposal at the end of the paper.

Round 2

Reviewer 1 Report

The biggest problem is the background of tree leave objects: are those backgrounds practically used in species identification. if single species are from various backgrounds, i can believe that there is not so big difference. authors should report, at least, how many different imaging sites (bacgrounds) have beeb used by species!
